# Integrating Geodesic Interpolation and Flow Matching for Non-Autoregressive Text Generation in Logit Space

## Abstract

Non-autoregressive language models are emerging as effective alternatives to autoregressive models in the field of natural language processing, facilitating simultaneous token generation. This study introduces a novel flow matching approach that employs Kullback-Leibler (KL) divergence geodesics to interpolate between initial and target distributions for discrete sequences. We formulate a loss function designed to maximize the conditional likelihood of discrete tokens and demonstrate that its maximizer corresponds to the flow matching velocity during logit interpolation. Although preliminary experiments conducted on the TinyStories dataset yielded suboptimal results, we propose an empirical sampling scheme based on a pretrained denoiser that significantly enhances performance. Additionally, we present a more general hybrid approach that achieves strong performance on more complex datasets, such as Fine Web and Lamini Instruction.

## 1 Introduction

Non-autoregressive language models have emerged as a promising alternative to traditional autoregressive models in natural language processing (NLP) tasks, potentially offering significant advantages during inference by enabling the simultaneous generation of all tokens rather than a sequential process. However, capturing the complex dependencies inherent in discrete textual data without relying on the autoregressive assumption presents substantial challenges.

In this paper, we investigate a conditional flow matching approach for text generation, which has garnered significant interest recently. Notably, methods such as discrete flow matching Gat et al. (2024) and Dirichlet flow matching Stärk et al. (2024) have adapted continuous flow-based models to address the discrete nature of text by representing tokens as one-hot vectors within a $V$-dimensional simplex, where $V$ denotes the vocabulary size.

The central concept in flow matching involves initiating a probability path $\rho_t$ between a known starting distribution $\rho_0$ and a target distribution $\rho_1$, which is only accessible through samples. In our context, these samples are provided as discrete sequences, represented as sequences of vectors within a $V$-dimensional simplex. Previous research has highlighted the limitations of linear interpolation in this framework Stärk et al. (2024). In response, we propose the utilization of geodesics under the Kullback-Leibler (KL) divergence, which effectively incorporates the inherent geometry of the simplex. These geodesics correspond to linear interpolation in the logit space, expressed as $l_t = \log x_t$, thereby facilitating a simple closed-form interpolation method.

The loss function within the generation CFM (Conditional Flow Matching) framework can be selected from various options; however, leveraging insights from recent studies, we implement a denoiser that maximizes the conditional likelihood $\log P_\theta(x_1|x_t, t)$, which predicts the discrete tokens of $x_1$. While it is feasible to model the joint conditional distribution, this approach does not confer any advantages over existing GPT models, as setting $t = 0$ necessitates modeling the unconditional data distribution. Therefore, we focus on modeling the **marginal likelihood** of each token in $x_1$ conditioned on $x_t$, which corresponds to a simplified approximation of the joint distribution. Although the theoretical justification for this loss function has been established in the context of single-token scenarios, its applicability to general sequences has not been comprehensively investigated. We provide a theoretical analysis establishing that the exact maximizer of the likelihood yields a precise

expression for the flow matching velocity under logit interpolation, thus offering a robust foundation for our training scheme.

Despite the theoretical rigor of our approach, initial experiments conducted on the TinyStories dataset Eldan & Li (2023) indicated that the model's performance was suboptimal. To address this issue, we propose an innovative empirical sampling scheme based on the same pretrained denoiser. In this scheme, given a specific time $t$ and a sample $x_t$, we sample $x_1$ from the distribution $P(x_1|x_t)$ and introduce additional noise to generate a sample $x_{t+h}$. This process is iteratively repeated to produce the final sequence. Although the theoretical foundations for this method are not yet completely established, our experimental results demonstrate substantial improvements compared to the initial approach.

Our contributions can be summarized as follows:

- We introduce the application of Kullback-Leibler (KL) divergence geodesics, which correspond to linear interpolation in the logit space, for flow matching in discrete sequence modeling.
- We present a theoretical justification demonstrating that the exact maximizer of the likelihood function yields the precise flow matching velocity in this context.
- We propose a novel empirical sampling scheme that, while currently lacking comprehensive theoretical justification, exhibits significant performance improvements in experiments conducted on complex datasets, including the Fine Web and Lamini Instruction datasets.

## 2 BACKGROUND

Flow matching Lipman et al. (2023) is a well-established method for determining a flow that connects samples from two distributions with densities $\rho_0$ and $\rho_1$. This is achieved by solving the continuity equation in relation to the time-dependent vector field $\overline{v}(x, t)$ and the time-dependent density $\rho(x, t)$, subject to the following boundary conditions:

$$\frac{\partial \rho(x, t)}{\partial t} = -\text{div}(\rho(x, t)\overline{v}(x, t)),$$
$$\rho(x, 0) = \rho_0(x), \quad \rho(x, 1) = \rho_1(x).$$

The function $\rho(x, t)$ is referred to as the probability density path. Typically, the initial distribution $\rho_0$ is chosen for convenience, often as a standard normal distribution, $\rho(x) = \mathcal{N}(0, I)$. The target distribution $\rho_1$ is unknown; we only possess a set of samples drawn from this distribution. Consequently, our objective is to approximate the vector field $v(x, t) \approx \overline{v}(x, t)$ using these available samples.

Starting from a defined vector field, we can construct a flow $x_t$, which represents a time-dependent map satisfying the ordinary differential equation (ODE):

$$\frac{\partial x_t}{\partial t} = v(x_t, t)$$

with the initial condition

$$x_{t=0} = x_0.$$

From this process, one can sample a point $x_0$ from the distribution $\rho_0$ and subsequently use the ODE to obtain a point $x_{t=1} = x_1$, which will have a distribution approximately equal to $\rho_1$. While the vector field or path solutions are not unique for given boundaries $\rho_0$ and $\rho_1$, the identification of any solution facilitates the sampling process from the unknown density $\rho_1$.

The computation of the vector field is typically done by minimizing the flow-matching objective Lipman et al. (2023), where $v_\theta(x_t, t)$ is a parametrized velocity

$$\mathcal{L}_{CFM}(\theta) = \mathbb{E}_t \mathbb{E}_{x_1, x_0} \|v_\theta(x_t(x_0, x_1), t) - x_t'\|^2, \quad (1)$$

where $x_t(x_0, x_1)$ is some (known) flow between $x_0$ and $x_1$ connecting $x_0$ and $x_1$ (for example, one can take $x_t = (1 - t)x_0 + tx_1$ in the simplest case). Here the dash indicates the time derivative. Time variable $t$ is uniformly distributed: $t \sim \mathcal{U}[0, 1]$ and random variables $x_0$ and $x_1$ are distributed according to the initial and final distributions, respectively: $x_0 \sim \rho_0, x_1 \sim \rho_1$.

## 2.1 Conditional flow matching for discrete sequences

In the context of discrete sequences, the variable $x_1$ assumes a finite set of values denoted as $\{x_1^{(k)}\}_{k=1}^N$. Nonetheless, we can conceptualize $x_1$ as a continuous quantity by interpreting its probabilities. Specifically, for a sequence of length 1, $x_1$ can be represented by a one-hot vector of the form $(0, \ldots, 1, \ldots, 0)$, where the entry at position $k$ corresponds to the selected value $x_1^{(k)}$ and all other entries are zero. This framework allows us to view the possible values of $x_1$ as the vertices of an $N$-dimensional simplex.

For the initial distribution $\rho_0$, it is appropriate to employ the Dirichlet distribution, which expresses a uniform distribution over the simplex. Therefore, the task of flow matching consists of identifying a random point on the simplex, denoted as $x_0$, and matching it to a particular vertex $k$ of the simplex.

To establish a flow on the simplex, we propose the implementation of geodesic interpolation, formulated as follows:

$$x_t(x_0, x_1) = \text{Softmax}\big((1 - t)\log(x_0) + t\log(x_1)\big) \tag{2}$$

where $t$ represents the interpolation parameter. Since $x_1$ is represented as a one-hot vector, we smooth this vector $x_1^s$ for the stability of the calculation $\log(x_1^s)$ by adding a constant component weighted by $\beta$:

$$x_1^s = (1 - \beta)x_1 + \frac{\beta}{N}\mathbf{1} \tag{3}$$

The vector field for geodesic interpolation can be expressed clearly as follows:

$$\frac{dx_t}{dt} = M(x_t)(\log(x_1) - \log(x_0)) \tag{4}$$

where $M(x_t) = I - x_t x_t^T$ represents the projection matrix. Since the previous formulas utilize the logit representation of the probabilities, we will adopt the following notations moving forward: $l_0 = \log(x_0)$, $l_t = \log(x_t)$, $l_1 = \log(x_1)$. Consequently, the vector field for addressing the flow matching problem in log space can be expressed as:

$$\frac{dl_t}{dt} = l_1 - l_0 \tag{5}$$

It is noteworthy that geodesic interpolation on the simplex for the flow matching problem results in linear flow matching in the logit space.

## 2.2 Denoising objective

In the context of geodesic interpolation in logits, as described in Equation 2, the standard flow matching (FM) problem outlined in Equation 1 can be reformulated as follows:

$$\mathcal{L}_{CFM}(\theta) = \mathbb{E}_{t,x_0,x_1}\|v_\theta(x_t, t) - (l_1 - l_0)\|^2 \tag{6}$$

By employing a variable substitution, we can express the velocity $\frac{dl_t}{dt}$ as a function that depends solely on $l_1$ and $l_t$:

$$l_1 - l_0 = \frac{l_1 - l_t}{1 - t} \tag{7}$$

In a similar manner, the variable $l_t$ can be eliminated from the loss functional through the reparameterization of the learnable function $v_\theta(x_t, t)$:

$$v_\theta(x_t, t) = \frac{\hat{v}_\theta(x_t, t) - l_t}{1 - t} \tag{8}$$

Consequently, the loss function is now solely reliant on the values of $x_t$ and $l_1$, which facilitates a modification in the averaging variables:

$$\mathcal{L}_{CFM}(\theta) = \mathbb{E}_{t,x_0,x_1}\|\hat{v}_\theta(x_t, t) - l_1\|^2 \tag{9}$$

Given that the probability $p_t(x_t)$ is entirely characterized by the expression $p(x_t \mid x_0, x_1)p_0(x_0)p_1(x_1)$, and since the loss functional is independent of the variable $x_0$, we can express the relationship as follows:

$$\mathbb{E}_{t,x_0,x_1} = \mathbb{E}_{t,x_t \sim p_t(x_t), x_1 \sim p(x_1|x_t)} \tag{10}$$

---

**Algorithm 1** Inference scheme (ODE)

---

**Require:** Initial distribution $p_0$; model parameterizing $p(x_1|x_t)$; parameter $K$ (number of iterations); parameter $h$ (time step size, default $1/K$).
1: Set $t = 0$
2: Sample $x_t \sim p_0$
3: **for** $i = 1$ to $K$ **do**
4:     Compute $w = \text{Softmax}(p(x_1|x_t))$
5:     Compute smoothed mean logit $\bar{l}_1 = w \log\left(1 - \beta + \frac{\beta}{N}\right) + (1 - w)\log\left(\frac{\beta}{N}\right)$
6:     Compute $l_t \leftarrow l_t + \frac{h}{1-t}(\bar{l}_1 - l_t)$
7:     Compute $x_t = \text{Softmax}(l_t)$
8:     Update $t \leftarrow t + h$
9: **end for**
10: **Return** $x_t$

---

Utilizing the relationship $p(x_t|x_1)p(x_1) = p(x_1|x_t)p(x_t)$, we can derive the following expression for the functional of the flow matching problem:

$$\mathcal{L}_{CFM}(\theta) = \mathbb{E}_{t,x_t \sim p_t(x_t), x_1 \sim p(x_1|x_t)} \|\hat{v}_\theta(x_t, t) - l_1\|^2 \tag{11}$$

The minimizer of this problem assumes a straightforward form by defining the optimal target point $\hat{v}(x_t, t)$ for the flow as the average of all potential target points $l_1$ associated with the flows that traverse the current point $x_t$:

$$\hat{v}(x_t, t) = \mathbb{E}_{x_1 \sim p(x_1|x_t)} l_1 \tag{12}$$

It is important to highlight that $l_1$ is structured as a matrix of the form $l_1 = (l_1^{(1)}, \ldots, l_1^{(S)})$. Each element $l_1^{(k)}$ corresponds to the target value of token $k$ in a sequence of length $S$. Since each $l_1^{(k)}$ operates independently of the others, we have the relationship $\int p(x_1|x_t)dx_1^{(1)} \ldots dx_1^{(k-1)}dx_1^{(k+1)} \ldots dx_1^{(S)} = p(x_1^{(k)}|x_t)$. This allows us to **take the expectation of each token independently of the others**:

$$\mathbb{E}_{x_1 \sim p(x_1|x_t)} l_1^{(k)} = \int l_1^{(k)} p(x_1|x_t) dx_1^{(1)} \ldots dx_1^{(S)} = \int l_1^{(k)} p(x_1^{(k)}|x_t) dx_1^{(k)} = E_{x_1^{(k)} \sim p(x_1^{(k)}|x_t)} l_1^{(k)} \tag{13}$$

Therefore, the final solution can be expressed in the following matrix form:

$$\hat{v}(x_t, t) = \left( \mathbb{E}_{x_1^{(1)} \sim p(x_1^{(1)}|x_t)} l_1^{(1)} \quad \ldots \quad \mathbb{E}_{x_1^{(S)} \sim p(x_1^{(S)}|x_t)} l_1^{(S)} \right) \tag{14}$$

Instead of directly predicting the target point $\hat{v}(x, t)$, we can optimize the conditional probabilities $p_\theta(x_1^{(k)}|x)$ and obtain the vector field by computing the expectation as specified in Equation 14. The denoising objective for the Conditional Flow Matching (DCFM) problem is defined as follows:

$$\mathcal{L}_{DCFM}(\theta) = -\mathbb{E}_{t,x_t \sim p_t(x_t)} \sum_{k=1}^{S} \mathbb{E}_{x_1^{(k)} \sim p(x_1^{(k)}|x_t)} \log p_\theta(x_1^{(k)}|x_t), \tag{15}$$

## 2.3 INFERENCE: ITERATIVE SAMPLING SCHEME

**ODE** In the standard formulation of flow matching, inference is performed by solving an ODE. For the denoising problem situated on a discrete manifold, it is essential to solve the following equation:

$$\frac{dl_t}{dt} = \frac{1}{1-t}\left(\mathbb{E}_{x_1 \sim p_\theta(x_1|x_t)} l_1 - l_t\right) \tag{16}$$

The denoiser model $\hat{v}_\theta(x_t, t)$ predicts the unnormalized probability $p(x_1^{(k)}|x_t)$ of the $k$-th token, up to a normalization constant, which denotes the likelihood of the $k$-th component of the current sample $x_t$ transitioning to a specific vertex of the simplex. It is noteworthy that in the interpolation

---

**Algorithm 2** Inference scheme (Randomized)

---

**Require:** Initial distribution $p_0$; model parameterizing $p(x_1|x_t)$; parameter $K$ (number of iterations); parameter $h$ (time step size, default $1/K$).
1: Set $t = 0$
2: Sample $x_t \sim p_0$
3: **for** $i = 1$ to $K$ **do**
4:     Sample $x_1 \sim p(x_1|x_t)$
5:     Sample $x_0 \sim p_0$
6:     Compute $l_t = (1 - t - h)\log(x_0) + (t + h)\log(x_1)$
7:     Compute $x_t = \text{Softmax}(l_t)$
8:     Update $t \leftarrow t + h$
9: **end for**
10: **Return** $x_t$

---

formula, the target logit is smoothed according to $l_1 = \log\left((1-\beta)x_1 + \frac{\beta}{N}\mathbf{1}\right)$. This smoothing introduces a correction to the value of the vector field $\frac{dl_t}{dt}$:

$$\frac{dl_t}{dt} = \frac{w}{1-t}\log\left(1 - \beta + \frac{\beta}{N}\right) + \frac{1-w}{1-t}\log\left(\frac{\beta}{N}\right) - \frac{l_t}{1-t} \tag{17}$$

where $w = \text{Softmax}(p(x_1|x_t))$. A detailed description of the procedure can be found in Algorithm 1. In practice, the ODE solution is effective primarily for low-dimensional problems. To address this, we propose a variant of the method that substitutes the exact mean $\mathbb{E}_{x_1 \sim p_\theta(x_1|x_t)} l_1$ with a one-sample estimate, given by $l_1 = \log(x_1)$ where $x_1 \sim p_\theta(x_1|x_t)$. In our experiments we called that approach Logit-FM (semi-randomized).

**Randomized**    A fundamentally new approach to inference involves sampling $x_{t+h}$ conditionally based on $x_t$. Theoretically, the conditional probability distribution $p(x_{t+h}|x_t)$ can be derived directly if we have knowledge of $p(x_1|x_t)$:

$$p(x_{t+h}|x_t) = \int p(x_{t+h}|x_1)p(x_1|x_t)\,dx_1.$$

Here, the term $p(x_{t+h}|x_1)$ can be expressed as:

$$p(x_{t+h}|x_1) = \int p(x_{t+h}|x_0, x_1)p(x_0)\,dx_0.$$

In practice, to sample from $p(x_{t+h}|x_t)$, the following steps are performed:

- Sample noise $x_0 \sim \rho_0$.
- Sample the target point $x_1$ conditioned on $x_t$: $x_1 \sim p(x_1|x_t)$.
- Compute the interpolation between the noise and target points at the shifted time $t + h$ using the relation:

$$x_{t+h} = \text{Softmax}\big((1 - t - h)\log x_0 + (t + h)\log x_1\big).$$

This procedure is further delineated in Algorithm 2. A critical limitation of this approach is the inability to establish an exact expression for $p(x_1|x_t)$. When analyzing a sequence comprised of two tokens, we can express the conditional probability as:

$$p(x_1|x_t) = p(x_1^{(1)}, x_1^{(2)}|x_t) = p(x_1^{(1)}|x_t) \cdot p(x_1^{(2)}|x_1^{(1)}, x_t),$$

where the superscript denotes the position of each token within the sequence. It is important to note that the denoising objective defined in equation equation 15 approximates the total probability $p(x_1^{(2)}|x_t)$, rather than the conditional probability $p(x_1^{(2)}|x_1^{(1)}, x_t)$. At time $t = 0$, we encounter a particularly unfavorable scenario:

$$p(x_1^{(2)}|x_{t=0}) = p(x_1^{(2)}),$$

indicating that the model effectively approximates the marginal distribution of the tokens, neglecting any potential dependencies among them. This challenge is partially mitigated by incorporating noise $x_0$ into the sample $x_1$ at time $t$ to derive a sample at time $t + h$. Nonetheless, there are no guarantees that this iterative sampling procedure will converge to the target distribution $\rho_1$. Interestingly, for simple data distributions, this technique has been shown to perform effectively, as demonstrated in the experimental section.

**Hybrid** Randomized inference demonstrates effectiveness for simple data distributions; however, its performance deteriorates when applied to more complex distributions, as evidenced by experimental results. To address this limitation, we propose a hybrid inference scheme. Our previous observations indicated that the randomized procedure exhibits its most significant shortcomings when the time $t$ is close to zero. Specifically, when sampling $x_1$ from the distribution $p(x_1|x_t)$, the calculation for $x_t$ is given by:

$$x_t = \text{Softmax}\big((1 - t)l_0 + tl_1\big),$$

which predominantly yields noise $l_0$ for sufficiently small values of $t$. To mitigate this issue, we recommend performing a series of ODE integration steps until $t$ reaches a specified threshold value, denoted as $t^*$. Once this threshold is surpassed, the procedure can transition to randomized sampling steps. This approach aims to enhance the robustness of inference in the face of complex data distributions.

## 3 RELATED WORK

In this section, we examine the existing literature on modeling discrete sequences. The authors in Campbell et al. (2024) introduce Discrete Flow Models (DFMs), which integrate discrete and continuous data through the use of Continuous Time Markov Chains. These models enhance traditional diffusion methods, facilitating multimodal frameworks for protein co-design and achieving state-of-the-art results in the generation of protein structures and sequences.

Moreover, Song et al. (2021) propose a stochastic differential equation (SDE) aimed at transforming complex data distributions by systematically adding and removing noise, leveraging neural networks for accurate score estimation. In addition, the work by Campbell et al. (2022) presents a continuous time framework specifically designed for denoising diffusion models of discrete data, resulting in the development of high-performance samplers that surpass conventional methodologies.

The research conducted by Gat et al. (2024) introduces Discrete Flow Matching, a novel approach focused on generating high-dimensional discrete data, such as language, while emphasizing improvements in generative perplexity. Additionally, Ghazvininejad et al. (2019b) utilize masked language modeling techniques to predict target words based on existing input text, and Austin et al. (2021) enhance multinomial diffusion models by incorporating transition matrices tailored for discrete data. Lastly, Hoogeboom et al. (2021) offer extensions to generative flows and diffusion models specifically for categorical data, demonstrating superior efficacy in text modeling and image segmentation applications.

Recent advancements have also concentrated on the application of continuous space diffusion methods to discrete datasets Dieleman et al. (2022); Li et al. (2022); Han et al. (2022). Noteworthy contributions by Lin et al. (2023) aim to refine the modeling of diffusion flows, while novel techniques for Continuous Flow Matching have been introduced by Lovelace et al. (2022) and Stärk et al. (2024) utilizing Dirichlet paths.

Autoregressive models have proven to be fundamental in the advancement of natural language processing Zhao et al. (2023). The influential GPT-2 model Radford et al. (2019) exemplified the potential of autoregressive approaches in producing coherent and contextually relevant text, thereby setting the groundwork for significant progress in a wide range of applications. Subsequent research has further explored these generative capabilities, underscoring the ability of autoregressive methodologies to tackle complex linguistic challenges.

Masked generative modeling has emerged as a promising area, utilizing various techniques to generate content by obscuring portions of the input data and predicting the masked variables Ghazvininejad et al. (2019a). Research conducted by Savinov et al. (2022) has investigated enhancements to

traditional masking techniques within the domain of text generation, leading to developments such as MaskGIT, which employs advanced methodologies for high-resolution image synthesis Chang et al. (2022). Furthermore, Ziv et al. (2024) illustrated the effectiveness of a text-to-music model, demonstrating that the adoption of the MaskGIT framework significantly elevates the quality of generated outputs.

# 4 EXPERIMENTS

In this study, we undertaken a comprehensive evaluation of our proposed method across both language modeling and image generation tasks. For the language modeling component, we compared our method to existing approaches by employing the established generative metric known as perplexity. In the context of the image generation task, we utilized the Frechet Inception Distance (FID) as our primary evaluation metric. All experiments were conducted utilizing two H100 GPUs, each equipped with 80 GB of memory.

## 4.1 EXPERIMENTAL SETUP

**Data**  For our experiments, we conducted evaluations using the Tiny Stories Eldan & Li (2023) and Binarized MNIST LeCun et al. (2010) datasets. The Tiny Stories dataset comprises synthetically generated short narratives produced by the models GPT-3.5 and GPT-4. This dataset employs a limited vocabulary, which significantly reduces the computational resources required for experiments. The Binarized MNIST dataset consists of black-and-white images of handwritten digits, each with a size of $28 \times 28$ pixels. This can be interpreted as a modeling problem involving sequences of length 784 and a vocabulary size of 2.

Furthermore, we extended our evaluation to more complex datasets, including the Fine Web Penedo et al. (2024) dataset, which contains over 15 trillion tokens of cleaned and deduplicated English web data. For our training purposes, we utilized a smaller subset of this dataset comprising 10 billion tokens, represented as sequences with a length of 1024 and a vocabulary size of $50, 257$.

Additionally, we assessed the proposed technique in the context of a conditional generation task, utilizing the Lamini Instruction Wu et al. (2023) dataset, which includes 2.58 million pairs of instructions and responses, each with a length of 256 tokens. In all experimental tasks, we applied a smoothing coefficient of $\beta = 0.01$ to facilitate geodesic interpolation.

**Model**  In our experiments, the architecture utilized for the toy tasks is based on the DiT model Peebles & Xie (2022). We conducted extensive experiments on high-dimensional data sets (Fine Web and Lamini Instruction), employing an efficient open-access implementation of GPT-2 [1], which was specifically modified to address the flow matching problem (see Appendix A). The text modeling architecture consists of 12 layers, 12 attention heads (6 heads for high-dimensional tasks), an embedding size of 768, and a vocabulary size of 50,257. For the image modeling task, we implemented a model characterized by 4 layers, 6 attention heads, an embedding size of 96, and a vocabulary size of 2.

**Evaluation**  The performance of unconditional text modeling was evaluated using the generative perplexity metric, which quantifies how well a language model predicts a sequence of words. This assessment was conducted with several advanced language models, including GPT-2 Radford et al. (2019), GPT-3 Brown et al. (2020), and Llama 2 Touvron et al. (2023). To further analyze the diversity of the generated texts, we employed the entropy metric, which measures the unpredictability or randomness in a dataset; a higher entropy value indicates greater diversity.

For the evaluation of conditional text generation, we utilized standard metrics including ROUGE-L, Bert-Score, and BLEU Score. ROUGE-L Lin (2004) measures the overlap between the generated text and reference text, focusing on recall and allowing for the assessment of longer sequences by identifying the longest common subsequence. Bert-Score Zhang* et al. (2020) leverages contextual embeddings from the BERT model to evaluate the semantic similarity between generated and reference texts, providing insight into both precision and recall of n-grams. BLEU Score Papineni et al.

---

[1]Github https://github.com/KellerJordan/modded-nanogpt

Table 1: Generative perplexity on unconditional text generation compared to prior work. Models were trained on Tiny Stories dataset.

| Method | NFE | Llama 2 | GPT 3 | GPT 2 |
|---|---|---|---|---|
| Data | - | 5.4 | 6.8 | 12.3 |
| Autoregressive | 128 | 9.1 | 9.9 | 14.8 |
| DFM | 32/64/128 | 24.5/21.5/23.1 | 28.4/24.3/26.3 | 34.3/29.5/31.5 |
| Logit-FM (randomized) | 32/64/128 | **8.0/7.4/6.8** | **8.2/7.4/6.7** | **11.6/10.8/10.0** |
| Logit-FM (semi-randomized) | 32/64/128 | 20.0/18.7/17.3 | 22.3/20.7/19.3 | 28.1/26.6/24.8 |
| Logit-FM (ODE) | 32/64/128 | 46.2/41.9/42.8 | 67.3/60.1/55.2 | 83.0/75.9/68.5 |

Table 2: Generative perplexity on unconditional text generation compared to prior work. Models were trained on Fine Web dataset.

| Method | NFE | Llama 2 | GPT 3 | GPT 2 |
|---|---|---|---|---|
| Data | - | 10.7 | 17.1 | 33.9 |
| Autoregressive | 1024 | 70.8 | 114.4 | 113.8 |
| DFM | 256/512/1024 | 206.2/159.5/110.7 | 353.5/253.8/156.6 | 390.8/278.8/172.4 |
| Logit-FM (ODE) | 256/512/1024 | 220.1/214.2/207.1 | 682.1/604.6/598.0 | 744.6/660.7/657.7 |
| Logit-FM (hybrid) | 256/512/1024 | 104.0/75.6/**54.5** | 159.9/**108.4**/73.5 | 180.2/121.5/**82.2** |

(2002) calculates the similarity between the generated text and reference text based on n-gram overlap, serving as a widely recognized metric for evaluating the quality of machine translation and text generation. Each of these metrics contributes to a comprehensive understanding of the performance and diversity of the generated texts.

## 4.2 UNCONDITIONAL LANGUAGE MODELING

We conducted a comparative analysis of our proposed method against established approaches, including autoregressive text generation utilizing a trained GPT-like model and a non-autoregressive model introduced in the Discrete Flow Matching (DFM) paper Gat et al. (2024). To ensure experimental rigor, we utilized the most basic version of DFM. The results of this comparison for toy experiments conducted on the Tiny Stories dataset are summarized in Table 1.

The observed results indicate that the performance of the trained Logit-FM model is highly dependent on the inference method employed. Specifically, the ODE inference method yielded the lowest text quality across all evaluation metrics. In contrast, both the DFM model and the semi-randomized Logit-FM model demonstrated comparable text quality, outperforming the autoregressive approach. Notably, the Logit-FM model operating under the randomized inference method produced output that closely matched real texts and significantly surpassed the autoregressive model in quality. Furthermore, we measured the entropy scores across all models and inference methods presented, finding that they exceeded a threshold of 5, which is indicative of diverse text generation.

Importantly, even with a substantial reduction in the number of function evaluations (NFE) to 32 and 64, the quality of the generated text, as assessed by perplexity, remained relatively high. These results were achieved using top-k sampling with $k = 1$ for the Logit-FM (randomized) method.

As illustrated in Figure 1, we investigated the effects of varying $k$ on text quality (perplexity) and variability (entropy). Our findings reveal that as $k$ increases, there is a slight increase in variability; however, a significant decline in quality is observed when $k$ exceeds 5. Nevertheless, even at $k = 1$, the text variability—measured by entropy—remained comparable to that of actual texts.

In addition to our previous analysis, we extended our investigation to include more complex datasets, specifically the Fine Web dataset. Our experiments with sequences of length 1024 revealed that the Logit-FM model consistently outperformed both the autoregressive and DFM approaches, as illus-

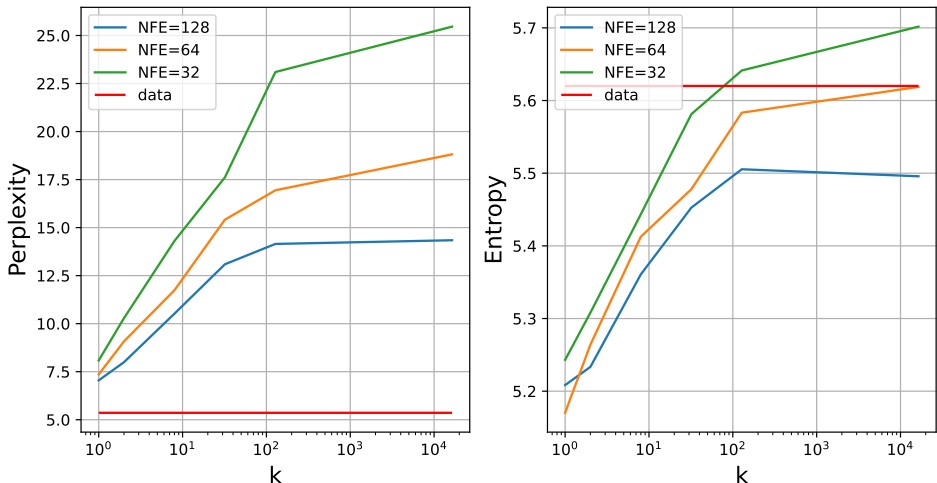

Figure 1: quantitative assessment of the impact of selecting the $k$ parameter for top-k sampling from the conditional probability $p(x_1|x_t)$ on the quality of the generated texts, measured in terms of perplexity, as well as their variability, assessed through entropy. For reference, the graphs include numerical estimates of the quality and variability of real texts, represented by a horizontal "data" line.

Table 3: Evaluation of conditional text generation compared to prior work.

| Method | ROUGE-L | BERT Score | BLEU Score |
|---|---|---|---|
| Autoregressive | 0.30 | 0.59 | 0.17 |
| DFM | 0.17 | 0.50 | 0.04 |
| Logit-FM (ODE) | 0.22 | 0.54 | 0.05 |
| Logit-FM (hybrid) | 0.24 | 0.57 | 0.07 |

trated in Table 2. The fully randomized approach (Logit-FM randomized) was excluded from the table due to a low entropy score of 3.7, which falls below the established threshold of 5 that indicates insufficient text diversity (with most tokens in the sequence being repeated). In contrast, hybrid approaches that integrate both ODE and randomized steps (see Appendix B) exhibited improved text quality (as measured by perplexity) for 1024 function evaluations, while also generating texts that satisfied the entropy requirement (exceeding 5). Notably, even with a twofold reduction in function evaluations, the hybrid method still produced texts of quality comparable to that of the autoregressive approach.

## 4.3 CONDITIONAL LANGUAGE MODELING

We have assessed the capability of the proposed methodology in addressing the conditional text generation problem. The evaluations were conducted using the Lamini Instruction dataset, which consists of sequences with total length of 512 (the maximal length of instruction and corresponding response). Performance metrics were analyzed based on several established evaluation criteria: ROUGE-L, BERT Score, and BLEU Score, as detailed in Table 3. Our findings indicate that the optimal balance between ODE and randomized steps in the hybrid (Logit-FM hybrid) approach occurs when the number of ODE steps is zero (see Appendix B), resulting in a fully randomized method. Consequently, no hybrid approach results are presented in the table. All values included in the table were computed as averages over 8 independent runs of response generation for each instruction. While the results for Logit-FM are slightly inferior to those of the autoregressive approach, they remain comparable. Furthermore, it was observed that both ODE and randomized approaches outperform the DFM approach across all evaluated metrics.

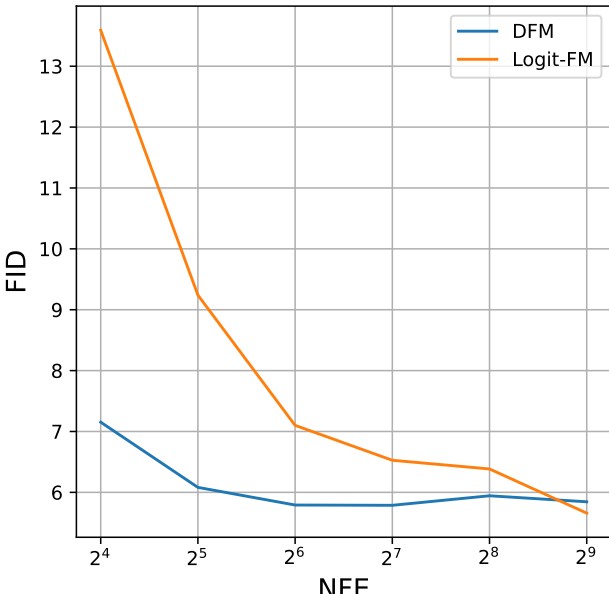

Figure 2: Quantitative comparison of the DFM and Logit-FM approaches in the context of image modeling task. The performance of each method is evaluated using the Frechet Inception Distance (FID) metric, plotted against the number of function evaluations (NFE).

## 4.4 IMAGE MODELING

We conducted an exploration into training the model to effectively reconstruct a picture manifold, using the Binarized MNIST dataset as a fundamental example. A comparative analysis of the proposed model, which utilizes (ODE) inference, against the diffusion probabilistic model (DFM) is presented in Figure 2. Our analysis indicates that for a number of function evaluations (NFE) less than 128, the DFM exhibits superior performance, as assessed by the Fréchet Inception Distance (FID) metric. The FID metric serves as a robust measure of the similarity between generated images and the real data distribution, thus providing insights into the quality of image generation. However, as the number of function evaluations increases beyond 128, we observe that the quality of images generated by both models begins to converge, suggesting that the enhancements provided by DFM diminish at higher NFE. This trend highlights the importance of balancing computational resources with the desired image quality in model training.

## 5 CONCLUSIONS AND FUTURE WORK

In this study, we introduced Logit-FM as a robust solution for discrete problems, employing geodesic interpolation within the simplex framework. We validated the use of a denoising objective, enabling the independent resolution of the problem for each token in the sequence. Additionally, we developed a novel inference scheme for the learned denoiser model, which demonstrates significant improvements over traditional inference methods in terms of the quality of generated sequences. Remarkably, our proposed method consistently outperforms autoregressive models in quality, even with a substantial reduction in the number of function evaluations.

Looking forward, we aim to establish a comprehensive theoretical foundation for our empirically effective inference scheme and to assess its performance across a wider array of complex tasks. We contend that the approach presented in this work has the potential to make a substantial contribution to the field of discrete data modeling, paving the way for future advancements and applications.

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

## A  OPTIMAL TRAINING CONFIGURATION

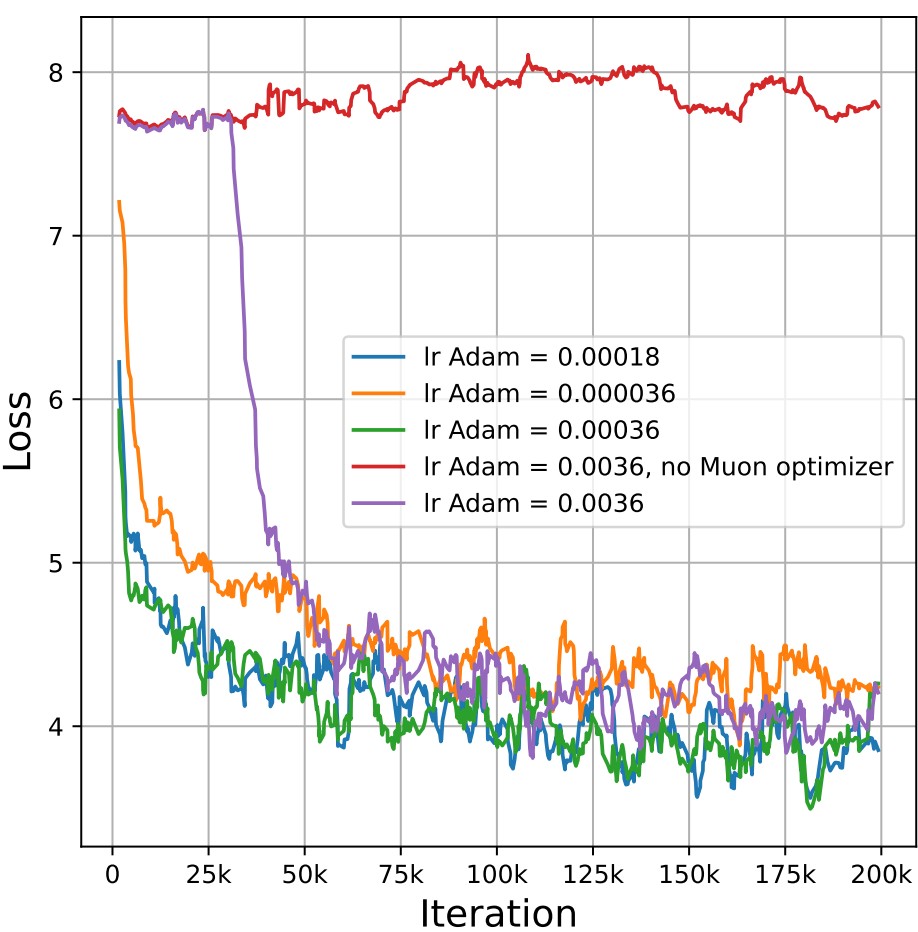

Figure 3: Comparison of the impact of learning rate values on training a GPT-like model for the Flow Matching problem. The base implementation utilizes the Muon optimizer for certain model parameters, while the tag "no Muon optimizer" indicates that the Muon optimizer has been replaced with the Adam optimizer.

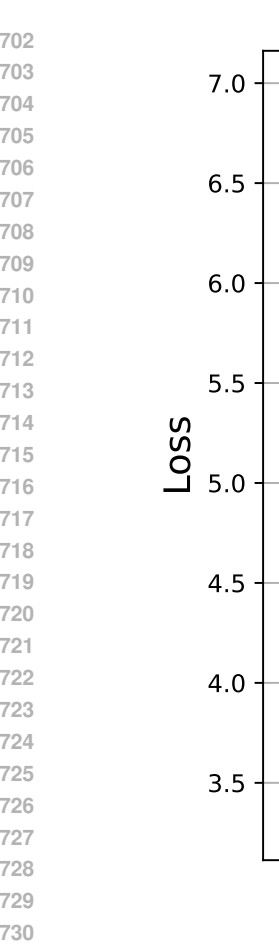

Figure 4: Comparison of various strategies for time insertion within model architecture.

In this section, we discuss several critical aspects and technical strategies for addressing the Flow Matching (FM) problem. The foundational code and architecture employed for training were derived from an open-source GitHub repository featuring an efficient implementation of the GPT-2 model, designed for standard language modeling tasks. However, our investigation revealed that the initially suggested optimal configuration is not truly optimal for the FM problem.

A key factor influencing convergence is the selection of an appropriate learning rate. In Figure 3, we present a comparison of various learning rate values, alongside an assessment of how the integration of the Muon optimizer—proposed in the original repository—affects model performance. We found that the standard learning rate of (lr = 0.0036) is not optimal. A learning rate reduced by a factor of ten significantly accelerates convergence and mitigates the risk of stagnation during the initial phases of training. Furthermore, we determined that the ratio of learning rates between the Adam optimizer and the Muon optimizer yields optimal results. Additionally, the application of the Muon optimizer for specific model parameters enhances convergence, even when employing a non-optimal learning rate.

Another critical consideration is the method of incorporating temporal information into the model architecture. We identified three primary strategies for this purpose:

- Time Token: Transform the time value into an embedding vector and incorporate it as a separate token within the sequence.
- Layer Normalization: Employ a method akin to that used in the DiT architecture, where the time embedding is utilized to adjust the mean and standard deviation of the data within the layer normalization module.

• Standard Addition: Simply append the time embedding to each token embedding.

Our findings, as presented in Figure 4, indicate that the Layer Normalization strategy is the most effective approach, as it provides better convergence and achieves a lower loss value after $200k$ training steps.

## B   OPTIMAL $t^*$ FOR HYBRID METHOD

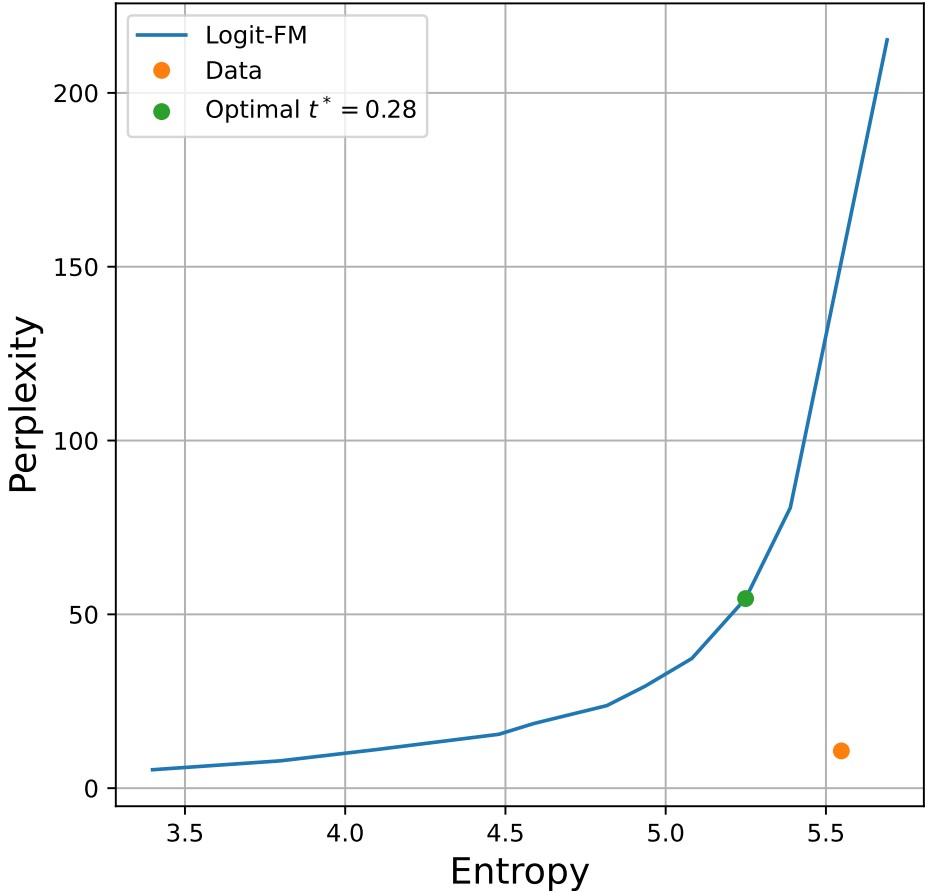

Figure 5: Comparison of the impact of various optimal splits $t^*$ (hybrid method) on entropy and perplexity, as computed using the Llama2 model.

In paragraph 2.3, we described the division of the inference procedure into two distinct phases. In the first phase, we perform ODE steps, while in the second phase, we implement randomized steps. The transition between these two phases is governed by the hyperparameter $t^*$. For the experiments summarized in Tables 2 and 3, we conducted a search to identify the optimal value of $t^*$. Figure 5 illustrates the influence of this choice on the entropy and perplexity of the generated texts. As indicated in the experimental section, the diverse texts exhibit an entropy value greater than 5. Therefore, the optimal $t^*$ must both meet this criterion and yield low perplexity. For the Fine Web dataset, we determined that the optimal value is $t^* = 0.28$.

For the Lamini Instruction dataset, we employed a similar methodology to ascertain the optimal $t^*$. The results are presented in Figure 6. Notably, the best scores for most metrics were achieved at $t^* = 0$, which corresponds to setting the number of ODE steps to zero. This finding indicates that the most effective inference strategy for this dataset is a fully randomized approach.

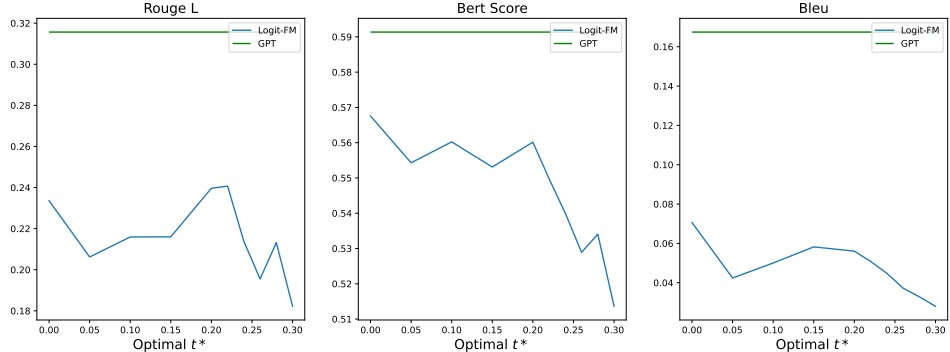

Figure 6: Comparison of the effects of different optimal splits $t^*$ (hybrid method) on conditional generation performance, as measured by ROUGE-L, BERT Score, and BLEU Score.

