# OpenReview forum: "Integrating Geodesic Interpolation and Flow Matching for Non-Autoregressive Text Generation in Logit Space"
_ICLR.cc/2025/Conference — Submitted to ICLR 2025_

### Official Review · Reviewer_94S6 · 2024-10-21

**Soundness:** 1
**Presentation:** 1
**Contribution:** 1
**Rating:** 1
**Confidence:** 2

**Summary:**

This work presents a flow matching approach for generating discrete sequences. This approach treats discrete tokens as one-hot vectors and constructs a flow by interpolation on the logit space. Randomized top-k sampling is proposed for inference.

**Strengths:**

N/A

**Weaknesses:**

This paper is only half-baked and needs substantial refinements before resubmission. For example, the presentation is poor (many variables are not explained, Figure 1/2 have the same caption, and some references are placeholders), experiments are only conducted on toy datasets (Tiny Stories, MNIST), and evaluation metrics are not sound (only use generative perplexity for language modeling).

**Questions:**

N/A

---

> ### Author Response · Authors · 2024-11-22
>
> **This paper is only half-baked and needs substantial refinements before resubmission. For example, the presentation is poor (many variables are not explained, Figure 1/2 have the same caption, and some references are placeholders), experiments are only conducted on toy datasets (Tiny Stories, MNIST), and evaluation metrics are not sound (only use generative perplexity for language modeling).**
>
> We apologize for any deficiencies in the writing quality of our manuscript. The complexity of the problem we addressed posed significant challenges in achieving optimal results, and we were under time constraints to complete the paper ahead of the submission deadline. We have addressed the identified issues and revised several sections of the text to enhance quality.
>
> We added two experiments described in general comment and paper revision. In the experiment conducted using the Lamini Instruction dataset, we utilized ROUGE-L, BERT Score, and BLEU Score to evaluate the quality of the generated responses in comparison to other methods. The results demonstrate that the performance of Logit-FM is comparable to that of the autoregressive approach and exceeds that of the DFM method.

---

### Official Review · Reviewer_CXED · 2024-10-28

**Soundness:** 2
**Presentation:** 3
**Contribution:** 2
**Rating:** 5
**Confidence:** 3

**Summary:**

The paper introduces a novel method for non-autoregressive text generation using KL-divergence geodesics and flow matching in logit space. The authors propose a conditional flow matching approach to address the challenges of discrete sequence modeling, demonstrating theoretical alignment between the loss function and flow matching velocity. To enhance performance, they implement an empirical sampling scheme based on a pretrained denoiser. Experiments on both text and image datasets show that the method outperforms traditional autoregressive models. Despite promising results, the sampling technique lacks full theoretical justification.

**Strengths:**

- The paper introduces a novel application of KL-divergence geodesics for text generation, addressing limitations in linear interpolation commonly encountered in discrete sequence modeling.

- The use of a pretrained denoiser-based empirical sampling scheme demonstrates ingenuity, compensating for initial performance shortcomings and achieving improved generation results.

**Weaknesses:**

- The paper seems to have been written in a hurry and lacks proper polish, with numerous missing references that make it difficult for me to follow. For example, references are missing at lines 32, 33, 39, 53, and 90, which disrupts the flow of the paper.
- I find the experimental section quite limited, as it only includes a single experiment for both text and image generation. A detailed ablation study is missing, making it hard to understand the impact of different components.
- I believe the evaluation metric for text generation is too restricted, relying almost exclusively on perplexity. While perplexity is useful for understanding how well the generated text fits the probable distribution, it can fail to capture semantic richness. I would recommend adding metrics like BLEU, ROUGE, or exploring newer evaluation methods for a more comprehensive assessment.
- After reading the introduction, I still do not fully understand why flow matching is necessary for generation models. The motivation for choosing this specific approach remains unclear to me.

**Questions:**

See the Weaknesses part

---

> ### Author Response · Authors · 2024-11-22
>
> **The paper seems to have been written in a hurry and lacks proper polish, with numerous missing references that make it difficult for me to follow. For example, references are missing at lines 32, 33, 39, 53, and 90, which disrupts the flow of the paper.**
>
> Thank you for your comment! We apologize for the deficiencies in the writing quality of our manuscript. The complexity of the problem we addressed posed significant challenges in achieving optimal results, and we were under time constraints to complete the paper ahead of the submission deadline. We have addressed the identified issues and revised several sections of the text to enhance quality.
>
> **I find the experimental section quite limited, as it only includes a single experiment for both text and image generation. A detailed ablation study is missing, making it hard to understand the impact of different components.**
>
> We added two experiments described in general comment.
>
>
> **I believe the evaluation metric for text generation is too restricted, relying almost exclusively on perplexity. While perplexity is useful for understanding how well the generated text fits the probable distribution, it can fail to capture semantic richness. I would recommend adding metrics like BLEU, ROUGE, or exploring newer evaluation methods for a more comprehensive assessment.**
>
> In the experiment conducted using the Lamini Instruction dataset, we utilized ROUGE-L, BERT Score, and BLEU Score to evaluate the quality of the generated responses in comparison to other methods. The results demonstrate that the performance of Logit-FM is comparable to that of the autoregressive approach and exceeds that of the DFM method.
>
> **After reading the introduction, I still do not fully understand why flow matching is necessary for generation models. The motivation for choosing this specific approach remains unclear to me.**
>
> The primary advantage of the flow matching (FM) technique lies in its flexibility to interpolate between source and target distributions. Therefore, we employ flow matching to integrate Kullback-Leibler (KL) divergence geodesics for discrete sequence modeling. Additionally, as discussed in the theoretical section, Kullback-Leibler divergence geodesics can be shown to be equivalent to linear flow matching in Logit space, which facilitates a clear derivation of the denoising objective.

---

### Official Review · Reviewer_UmBn · 2024-11-02

**Soundness:** 3
**Presentation:** 1
**Contribution:** 2
**Rating:** 3
**Confidence:** 3

**Summary:**

This paper presents a novel approach for non-autoregressive text generation in logit space. It uses Kullback-Leibler (KL) divergence geodesics for flow matching between initial and target distributions of discrete sequences. A loss function is defined to maximize the conditional likelihood of discrete tokens, and its theoretical properties are explored. Despite initial poor results on the TinyStories dataset, an empirical sampling scheme based on a pretrained denoiser is proposed, which significantly improves performance. The method is also applied to image generation tasks for comparison.

**Strengths:**

1) Novel theoretical approach: The use of KL-divergence geodesics for flow matching in discrete sequence modeling is a novel concept. The theoretical justification provided for the likelihood function and its relation to the flow matching velocity adds to the rigor of the method.

**Weaknesses:**

1) Extremely low writing quality: The writing and presentation of this article are extremely poor and unreasonable.

2) Limited dataset evaluation: The evaluation is conducted on two uncommon datasets.

**Questions:**

Given the low quality of presentation, I have no further questions. I hope that the authors can make full preparations and improvements before the next submission.

---

> ### Author Response · Authors · 2024-11-22
>
> **Extremely low writing quality: The writing and presentation of this article are extremely poor and unreasonable.**
>
> We apologize for any deficiencies in the writing quality of our manuscript. The complexity of the problem we addressed posed significant challenges in achieving optimal results, and we were under time constraints to complete the paper ahead of the submission deadline. We have addressed the identified issues and revised several sections of the text to enhance quality.
>
> **Limited dataset evaluation: The evaluation is conducted on two uncommon datasets.**
>
> We added two experiments described in general comment.

---

> > ### Comment · Reviewer_UmBn · 2024-11-22
> >
> > Received. Please allow me some time to go through your revised manuscript before giving you feedback.

---

### Author Response · Authors · 2024-11-22
**We conducted two additional experiments in which the Logit-FM method outperformed DFM and demonstrated comparable or superior results to the autoregressive model, enhancing the text in light of the new experiments and clarifying the mathematical motivation.**

We would like to express our gratitude to the reviewers for their thorough consideration of our research work. In response to their comments, we conducted two additional experiments that are detailed in the revised manuscript:

1) We trained a model on sequences of length 1024, utilizing 10 billion tokens from the Fine Web dataset, which consists of cleaned and deduplicated English web data (refer to Table 2 in the revision).

2) We also trained a model for the conditional language modeling task employing the Lamini Instruction dataset, comprised of 2.58 million pairs of instructions and responses, with a sequence length of 512 (see Table 3 in the revision).

In both experiments, the Logit-FM method consistently outperformed the DFM approach and demonstrated performance that is either superior to or comparable with that of the autoregressive model. For the evaluation of the model on the Lamini Instruction dataset, we utilized several metrics, including ROUGE-L, BERT Score, and BLEU Score.

The text has been revised to incorporate new experiments (Tables 2 and 3) and results analysis. Additionally, the mathematical motivation behind the randomized inference approach has been clarified. Furthermore, Appendices A and B include an investigation of the optimal FM training configuration and hyperparameters for the inference procedure.

---

### Meta-Review · Area_Chair_1ipN · 2024-12-19

**Metareview:**

The paper introduces a novel non-autoregressive text generation method using Kullback-Leibler (KL) divergence geodesics for flow matching in logit space. It proposes a conditional flow matching approach to address challenges in discrete sequence modeling and define a loss function to maximize the conditional likelihood of discrete tokens.  However, all reviewers consistently vote for rejection due to the paper seems  half-baked and should be further improved by carefully polishing in writing and adding more experiments to demonstrate their method.

**Additional Comments On Reviewer Discussion:**

The authors have conducted additional experiments in response to reviewer comments, enhancing the paper's credibility. However, this manuscript is still not ready for publishing on ICLR.

---

### Decision · Program_Chairs · 2025-01-22

Reject